# Factors associated with iron deficiency anaemia among pregnant teenagers in Ashanti Region, Ghana: A hospital-based prospective cohort study

Reginald Adjetey Annan[1]*, Linda Afriyie Gyimah[1], Charles Apprey[1], Anthony Kwaku Edusei[2], Odeafo Asamoah-Boakye[1], Linda Nana Esi Aduku[1‡], Wisdom Azanu[3‡], Herman E. Lutterodt[4‡]

1 Faculty of Biosciences, Department of Biochemistry and Biotechnology, College of Science, Kwame Nkrumah University of Science and Technology, Kumasi, Ghana, 2 Department of Community Health, School of Public Health, Kwame Nkrumah University of Science and Technology, Kumasi, Ghana, 3 Department of Obstetrics and Gynaecology, University of Allied Health Sciences, Ho, Ghana, 4 Department of Food Science and Technology, Kwame Nkrumah University of Science and Technology, Kumasi, Ghana

☯ These authors contributed equally to this work.
‡ These authors also contributed equally to this work
* reggie@imtf.org

**Data Availability Statement:** Data cannot be shared publicly because it contains sensitive identifying information. However, data are available

## Abstract

### Background

Iron Deficiency Anaemia (IDA) is reportedly high in pregnant adults and the causes well studied. However, among pregnant teenagers, the levels and associated factors of IDA are not fully understood.

### Methods

In a prospective cohort study among Ghanaian pregnant teenagers, aged 13–19 years, IDA prevalence and associated factors were investigated. Sociodemographic data, household hunger scale (HHS), lived poverty index (LPI), FAO's women's dietary diversity score (WDDS) and interventions received during antenatal care (ANC) were obtained from 416 pregnant teenagers in Ashanti Region, Ghana. Micronutrient intakes using a repeated 24-hour dietary recall, and mid-upper arm circumference (MUAC) were determined and blood samples analysed for haemoglobin (Hb), serum levels of ferritin, prealbumin, vitamin A, total antioxidant capacity (TAC), C-reactive protein (CRP), and zinc protoporphyrin (ZPP).

### Results

Anaemia (Hb cutoff <11.0 g/dL) was 57.1%; deficient systemic supply of iron stores (31.4%), depleted body stores of iron (4.4%), inadequate dietary iron intake (94.5%), and inadequate multiple micronutrient intakes (49.5%), were all notable among study participants. Between-subject effects using Generalized Linear Modelling indicated malaria tablet given at ANC (p = 0.035), MUAC (p = 0.043), ZPP (p<0.001), ZPP/Hb ratio (p<0.001) and

from the Committee on Human Research, Publication, and Ethics (CHRPE), the ethics board of the School of Medical Sciences of the Kwame Nkrumah University of Science and Technology, KNUST and Komfo Anokye Teaching Hospital (KATH), Kumasi, Ghana (Email: chrpe.knust. kath@gmail.com) for researchers who meet the criteria for access to confidential data.

**Funding:** Funding for this study was provided by the Nestle Foundation. The funders had no role in study design, data collection and analysis, decision to publish, or preparation of the manuscript.

**Competing interests:** The authors have declared that no competing interests existed regarding the publication of this study.

depleted body iron stores (DBIS) (p<0.001) to significantly affect Hb levels. Pregnant teenagers with a high ZPP/Hb ratio (OR = 9.7, p<0.001, 95%CI = 6.0–15.8) had increased odds of being anaemic compared to those with normal ZPP/Hb ratio. Participants who were wasted (OR = 1.2, p = 0.543, 95%CI = 0.6–2.3), and those with depleted iron stores (OR = 3.0, p = 0.167, 95%CI = 0.6–14.6) had increased odds of being anaemic. Participants who experienced hunger were close to 3 times more likely (OR = 2.9, p = 0.040, 95%CI = 1.1–7.8) for depleted iron stores, compared to those who did not experience hunger. Also, participants with inadequate multiple micronutrients intakes (OR = 2.6, p = 0.102, 95%CI = 0.8–8.4), and those with low serum levels of ferritin (OR = 3.3, p = 0.291, 95%CI = 0.4–29.2) had increased odds of depleted body iron stores.

## Conclusions

IDA is common among pregnant teenagers and the related factors include malaria tablets given at ANC, maternal hunger, maternal MUAC, a deficient systemic supply of iron, depleted body iron stores, ZPP, and ZPP/Hb ratio. Appropriate interventions are urgently needed to address the causes of IDA among pregnant teenagers.

## Introduction

Adolescence, a critical period of physical growth, early learning, and mental developmental changes have gained public interest and is being seen as a life state suited for strategic health policies and commitments [1,2]. Ghana has a youthful population, with the adolescent age group comprising about 22.0% of the total population [3]. Of all the health problems faced by adolescents, teenage pregnancy is arguable the most devastating due to its negative consequences on the health of the mother and infant [2]. Globally, it is estimated that 12 million girls aged 15–19 years have a child annually, and at least 770,000 of these childbirth takes place in developing countries [4]. The current Ghana demographic health survey shows that 14% of girls aged between 15 and 19 years had begun childbearing, with 11% live birth rate [5]. It is also known that early pregnancy and postnatal complications such as unsafe abortion, stillbirth, and pregnancy-induced hypertension are the main causes of mortality among girls aged 15–19 years old worldwide [6–8]. Iron deficiency anaemia is also a common complication that increases the risk of adverse pregnancy outcomes, such as low birth weight, preterm birth, perinatal and infant mortality, postpartum haemorrhage, and spontaneous abortion [9,10]. This implies that pregnant adolescent girls face a double burden of health risk because they have increased nutrients demand to ensure their own growth/development, in addition to that of their growing fetuses. Consequently, the UN's Sustainable Development Agenda on reducing poverty and hunger, promoting healthy lives and wellbeing at all ages, empowering women, and achieving equal education may not be attained if certain capacities of the adolescents are not developed [11].

Maternal anaemia has been a long-standing interest in the public health domain [12]. Among all the pathological causes of anaemia in pregnancy, iron deficiency anaemia is the most frequent, particularly in low-and-middle-income countries (LMICs), where the contribution from other anaemia disorders such as malaria and sickle cell disease are less significant [12–14]. Maternal anaemia occurs when the haemoglobin concentration is less than 11.0g/dL, while in the absence of inflammation, ferritin concentration less than 15 ug/L is also used to

determine iron-deficiency anaemia [15]. According to WHO, more than 40 percent of maternal women globally are anaemic due to iron deficiency [16], and/or folate and other micronutrient deficiencies [17,18]. Among the many health interventions including antenatal and postnatal maternal care services, the Community-based Health Planning and Services initiatives in poorly resourced communities, and iron-folic acid supplementation given to pregnant women to reduce the burden of anaemia [19], Ghana is still reporting a high prevalence of anaemia (45.1%) among pregnant women aged 15–49 [20]. A recent retrospective study by Ampiah et al. [21] also reported a very high prevalence of anaemia among adults and teenage pregnant women, higher among teenagers, and not much reduced over five years.

There are two forms of iron deficiency; absolute and functional. Absolute iron deficiency occurs as a result of low or depleted body stores of iron while, functional iron deficiency is a disorder that occurs when total body stores are normal or increased, but the iron supply to the bone marrow is deficient [22,23]. The two forms can coexist in pregnancy when risk factors are present [23]. The known causes of iron deficiency anaemia in pregnant women have been poor dietary intake as a result of food deprivation, consumption of less diverse foods, leading to poor nutritional status, and low iron stores, and therefore anaemia [24–27]. During pregnancy, there is an additional requirement for iron and folic acid to meet the pregnant woman's own nutritional needs as well as those of the developing foetus [16]. This is more so for the adolescent, who require nutrients for growth and development spurt experienced during this age as well. The reasons for the perpetual high anaemia prevalence are not obvious, but possibly, the underlying factors have not been fully investigated or understood. Researchers on anaemia have focused on adult pregnant women, and research interest into risk factors focused on related-causes such as; intestinal parasitic infections [28], malaria, HIV infections [29,30], sickle cell anaemia [31]. Other studies in Ghana have shown that multiparity, maternal underweight [32], poor knowledge about anaemia [33], malaria infection, no fish/meat consumption during pregnancy [34] are all predicting factors of anaemia among pregnant women that affect pregnancy outcomes.

Evidence from the above literature concludes that maternal anaemia has multifactorial causes, that negatively affect maternal and birth outcomes, but these are understudied among pregnant teenagers. The interactive effects of both immediate and underlying causes of anaemia: socio-demographic, antenatal intervention uptake, poverty and hunger, dietary factors, anthropometric and nutritional status on anaemia among Ghanaian pregnant teenagers are not known. Our study sought to investigate these factors to deepen our understanding of the determinants of anaemia during pregnancy.

## Methods

### Study setting

Teenage pregnant girls were recruited from 29 communities in Kumasi Metropolis, Asante Akim Central, Ejisu Juaben, Bosomtwi, Asante Akim South and North and Ahafo Ano North and South Districts, all in Ashanti Region, Ghana, from May to August 2018. Ashanti Region has an estimated population of 5,792,187, accounting for 19% of Ghana's total population in 2019 [35]. The report from the Ghana Maternal Health Survey 2017 showed a high prevalence of teenage pregnancy (12.2%) and anaemia in the region [36]. The study areas have hospitals, health centres, or Community-based Health Planning Services (CHPS) compounds. The study population consisted of pregnant teenagers (aged 13 to 19 years old) with gestational age up to 32 gestational weeks, who were attending antenatal clinics at their district health facilities.

## Study design

The study is an excerpt from a hospital-based prospective cohort study, involving 416 pregnant teenagers, aged 13 to 19 years old, residing in the 29 communities. The main study recruited and collected baseline data on pregnant teenagers. They were subsequently followed and birth outcomes evaluated upon delivery. The current study focuses on the baseline characteristics of the participants.

## Sample size calculation

This study is part of the larger longitudinal study which obtained birth outcome information of pregnant teenagers. Thus, the sample size used in this study was calculated from the larger study. Low birth weight was the main outcome in the larger study and was used in calculating the sample size. The sample size was calculated using the formula from Charan and Biswas study [37]:

$$n = 2(Z \alpha/2 + Z \beta)2 \, p(1-p)/(P1-P2)\hat{2}.$$

Where, n = sample size, Z $\alpha$/2 = 1.96 at type 1 error of 5%, Z $\beta$ = 0.84 at 80% power, P1 = LBW in pregnant adolescents with adequate nutritional status (11.6%), P2 = LBW in pregnant adolescents with poor nutritional status (23.3%), p1-p2 = difference in prevalence of low birth weight between pregnant adolescents with adequate nutritional status at birth and those with inadequate nutritional status, and p = pooled prevalence = (p1 +p2)/2. A pilot study among pregnant adolescents in the area reported LBW prevalence of 23.3% [38]. Hence, we proposed that LBW in pregnant adolescents with adequate nutritional status would be 11.4%, while those with poor nutritional status would remain 23.3%, a reduction of just above half (51.1%). Hence, p1 = 11.4%, p2 = 23.3%, their proportions being p1 = 0.114 and p2 = 0.233, and p = (0.114 +0.233)/2 = 0.1735.

Using the above descriptive, the sample size n = 2(1.96+0.84) 2 x 0.1735(1–0.233)/(0.114–0.233) 2, n = 2.09/0.01, equal 209 was calculated, which implied we needed to recruit 209 participants in each arm of the study (half in the poorly nourished group and a half in the well-nourished group) making 418 participants showing a significant association between poor nutrition and LBW. However, we added 10% attrition to give 460 participants who were needed. However, the study had 416 participants.

## Data sampling

A convenience sampling method using a first come first served approach was used in recruiting participants from hospitals, health centres, and CHPS zones during antenatal care visits. The inclusion criteria were that the pregnant teenagers were within the ages 13–19 years old, and also resided in the selected communities where the health centres/hospitals visited served. We obtained days in which antenatal clinics were held for pregnant adolescents in the hospitals/health centres involved in this study. On these dates, research assistants visited the hospitals/health centres, and any pregnant teenager within the required age group (13–19 years) was eligible for recruitment. Although special antenatal clinics were held for teenagers, attendance was low due to stigma in the rural districts. Hence, community information centres were asked to announce and invite all pregnant adolescents to the health centres. Those that came and consented to the study were recruited.

## Ethics

The study was approved by the Council for Scientific and Industrial Research (CSIR), Ghana (Reference: CHPRE/AP/236/18). Study protocols/aims were first explained to all participants

in their local language (Asante Twi). Written and signed informed consent was obtained from all participants by following CSIR protocols for recruiting for the study. Also, parents/guardians of the teenagers gave consent on behalf of participants less than 18 years old.

## Data collection

Data were collected on socio-demographic variables, dietary diversity, availability of food and poverty, and dietary intake. Anthropometric measurements were done and blood samples collected for haematology and biochemical analysis. Antenatal interventions uptake and other pregnancy-related practices were collected with a structured questionnaire. Data on age and parity were verified from their National Health Insurance Identification cards and maternal health record books of participants. A two-day training workshop was undertaken by the research team to train all enumerators on each data collection tool. A day field pre-testing session followed the training in a nearby community health centre. These trained enumerators carried out all data collection for this study at the health centres/hospitals. The trained enumerators consisted of MPhil Human Nutrition and Dietetics students, Phlebotomists and other field research assistants.

## Dietary assessment

A repeated 24-hour dietary recall on 2 weekdays and 1 weekend were used to obtain dietary intake of the participants, beginning from the previous day's dietary intake. All dietary recalls were done by face-to-face interviews by the trained Human Nutrition and Dietetics students, to cover intake for three days, including a weekend during the clinic visit. Household food measures were used to quantify food and beverages consumed by pregnant teenagers, and the portions in household measures were later converted into grams equivalents. The trained MPhil Human Nutrition students computed the grams of the food and beverages consumed by participants into a nutrient analysis Microsoft Excel software designed by the University of Ghana, Department of Food Science and Nutrition [39] which contains macro-and-micronutrients levels of Ghanaian foods. We compared the mean micronutrients obtained to the estimated average requirement (EAR) for pregnant women using the Institute of Medicine of the National Academy of Sciences cut-offs [40]. We considered micronutrient intakes below the EAR as "inadequate" and intakes equal to or above the EAR as "adequate" [40].

Besides, we also determined participants who had combined adequacies/inadequacies for iron, folate, and vitamin $B_{12}$ when compared with the EAR, and this was reported as multiple micronutrient intake (MMI). Participants who had three inadequate intakes for iron, folate, and vitamin $B_{12}$ were termed as inadequate multiple micronutrient intake (IMMI), inadequate for both iron-folate was classified as inadequate iron-folate intake, and inadequate for both iron-vitamin $B_{12}$ was termed as inadequate iron-vitamin $B_{12}$ intake.

Dietary diversity was assessed using a repeated 24-hour dietary recall method, taken for three days (two weekdays and one weekend). FAO's WDDS was then used to assess the dietary diversity of participants [41]. The WDDS consisted of ten (10) food groups, namely; grains, white roots and tubers, plantain; pulses (beans, pea, lentils); nuts and seeds; dairy; meat, poultry, and fish; eggs; dark green leafy vegetables; other vitamin A-rich fruits and vegetables; other vegetables; other fruits [41]. We used the first 24-hour recall from each participant to score her dietary diversity using the 10 food groups. The WDDS was then classified as low ($\leq$ 3), medium (4–5 food groups), or high ($\geq$ 6 food groups) according to FAO's guidelines [42]. The WDDS was further regrouped into adequate (5–9) and inadequate (0–4) dietary diversity scores to help with the statistical association for the population.

## Household hunger and poverty status assessment

The Household Hunger Scale designed by Ballard [43], was adopted to assess food availability and deprivation of pregnant teenagers while the Lived Poverty Index designed by Mattes [44] determining poverty status. The HHS assesses questions related to food availability and deprivation over the past month. The responses to these questions are coded and scored, ranging from 0–6. Scores less than 2 shows little or no hunger, 2–3 shows moderate hunger, and 4–6 shows severe hunger [43]. The LPI assessed questions related to the availability of food, water, cash income, medical care, and cooking fuel over the past year. The responses are scored on a five-point scale, ranging from 0 (which can be thought of as no lived poverty) to 4 which reflects a constant absence of all primary necessities [44]. The averages of the scores were classified as low (0–0.5), low-moderate (0.51–1.0), high-moderate (1.1–1.5) and high (> 1.5) LPI.

## Assessment of anthropometric status

The mid-upper arm circumference (MUAC) of participants was determined during the clinic visit. MUAC measurement was carried out by the trained enumerators who followed the recommended World Health Organization standard protocol. MUAC measurement was determined as a proxy indicator of maternal weight status since it has good specificity in determining weight during pregnancy [45,46]. An inelastic tape measure (Goldmoon body tape measure, Jiangsu, JS, China) was used to take MUAC of participants by locating the midpoint between the acromion and olecranon bone on the left hand of participants. MUAC measurement less than 22.0cm was referred to as severe wasting, 22.0cm to less than 24.0cm as mild/moderate wasting, and 24.0cm and above as normal MUAC [47]. MUAC assessment was measured twice and the average was used.

**Determination of haemoglobin (Hb) levels.** The trained phlebotomists took all blood samples at the health centres/hospitals, and samples were transported on dry ice to Kwame Nkrumah University of Science and Technology, Clinical Analysis Laboratory (CAnLab), Department of Biochemistry and Biotechnology for all the biochemical analyses. Two millilitres of venous blood sample of participants were collected into anticoagulant Ethylenediaminetetraacetic acid (EDTA) tubes and used to determine haemoglobin levels of participants using Sysmex Haematology system (XP-300 Automated Hematology Analyzer, Rhode Island, RI, USA). The WHO cut-off for Hb was used to determine anaemia prevalence among the participants; haemoglobin values less than 11.0g/dL means anaemia, 11.0g/dL and above means no anaemia [15].

**Assessment of serum nutrients.** All the serum nutrients were analysed following standardized laboratory protocols and performed by trained laboratory scientists. Serum ferritin is the main body storage of iron and is recommended by WHO in the assessment of iron levels, in the absence of inflammation [48,49]. However, serum ferritin is likely to be high (above 200 μg/L)) in the presence of inflammation or infection [49]. Thus, an inflammatory marker, CRP, was also assessed. In this study, ferritin levels were less than 200μg/L for all the participants, depicting no likelihood of inflammation or infection. Five millilitres venous blood taken from participants into serum gel separator tubes were centrifuged at 4000 rpm for 5 minutes to obtain the serum. The serum was used for the analysis of serum ferritin, prealbumin, c-reactive protein, total antioxidant capacity and zinc protoporphyrin using a sandwich enzyme-linked immunosorbent assay (ELISA) technique (from R&D system Inc, USA) at the Clinical Analysis Laboratory of the Department of Biochemistry and Biotechnology, Kwame Nkrumah University of Science and Technology. The optical density for serum ferritin, serum prealbumin, and serum ZPP was taken at 450 nm wavelength within 15 minutes, using a multipurpose microplate ELISA reader (Mindray MR-96A, Guangdong, China). TAC was determined using

the ferric reducing ability of plasma (FRAP) assay protocol described by Benzie and Strain [50]. TAC was measured at wavelength 593nm, with the help of a spectrophotometer (Mindray BA-88A, China). The standard curves of known concentrations of the respective recombinant value were used for the calculation of biochemical variables. All biochemical analyses were done in duplicates. Serum iron deficiency was termed as serum ferritin less than 15 μg/L [51]. Low serum prealbumin was defined as serum levels of less than 50–500 mg/L [52]. Serum CRP levels higher than 5–20 mg/L was termed high CRP [53]. ZPP values were measured as μg/dL and converted to μmol/mol heme (ZPP/Hb ratio) by calculating the following formula used by Stanton et al. [54] and Yu [55].

$$\text{ZPP/Hb ratio} = \frac{\mu\text{mol of ZPP}}{\text{mol of haemoglobin}} = \frac{\mu\text{g of ZPP}}{\text{gram of Hb}} \times 25.8$$

Deficient systemic iron supply was defined as ZPP/Hb ratio greater than 80 μmol/mol heme, ZPP/Hb ratio between 60–80 μmol/mol heme was termed moderate-high, and less than 60 μmol/mol heme means adequate systemic iron supply [56]. Depleted body iron stores were defined as participants with ZPP/Hb ratio greater than 80 μmol/mol heme, and serum ferritin less than 20μg/L [56].

### Data analysis

All data were analysed using IBM Statistical Package for Social Sciences version 25 (SPSS IBM Inc Chicago, USA). Absolute and relative frequencies were determined for sociodemography, hunger status, poverty status, dietary diversity, antenatal intervention uptake, micronutrient intakes, anthropometric, haemoglobin level, and biochemical variables. Kolmogorov-Smirnov test of normality was performed to determine whether all continuous variables met parametric assumptions. A chi-square (Fisher's exact test) cross-tabulation was performed to compare frequencies of hunger status, poverty status, women's dietary diversity score, antenatal intervention uptake, micronutrient intakes, anthropometric, haemoglobin level, biochemical variables, and iron deficiency anaemia status. An independent t-test and a two-way analysis of variance (ANOVA) (Generalized Linear Model test) were used for parametric comparisons, while Mann Whitney 'U' test was performed for non-parametric comparisons of all continuous variables. The mean (±Standard error mean) difference for the continuous variable; serum ferritin and prealbumin were compared by iron deficiency anaemia status, while the median(interquartile) differences for HHS, LPI, WDDS, MUAC and other biochemical variables were compared by iron deficiency anaemia status. Univariate tests of variables associations were performed using the generalized linear model test to determine the combined effect of study variables on haemoglobin level. Study variables that were significant at Chi-square/ANOVA analysis were put into the Binary logistic regression model to determine predictors of iron deficiency anaemia. All tests were 2-tailed, and p-values < 0.05 were termed significant.

## Results

Fig 1 indicates the prevalence of iron-deficiency anaemia among pregnant teenagers. Anaemia prevalence by haemoglobin status was 57.1%, and 31.4% of the participants had a poor systemic supply of body iron stores. Additionally, 4.4% of the pregnant teenagers were depleted of body stores of iron. In terms of dietary iron deficiency anaemia, 94.5%, of the participants had dietary intake below the EAR, while 82.5% and 54.3% of participants had combined dietary inadequate intakes for iron-folate, and iron-vitamin $B_{12}$, respectively. Also, 49.5% of the pregnant teenagerss had inadequate multiple micronutrient intakes (iron-folate-vitamin $B_{12}$ deficiencies) when compared with the EAR.

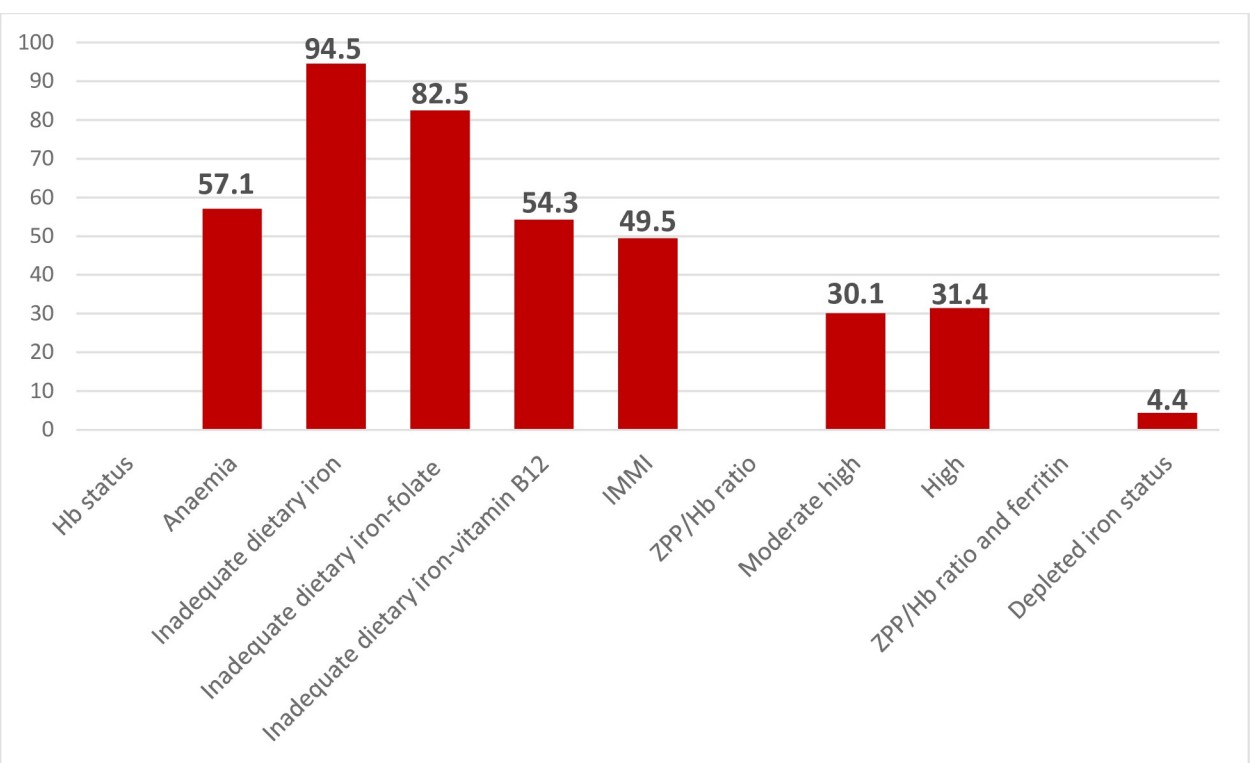

Hb- Haemoglobin, ZPP- Zinc protoporphyrin, IMMI- Inadequate Multiple Micronutrient Intakes for dietary iron, folate and vitamin B$_{12}$. Values are reported in percentage.

**Fig 1. Iron-deficiency anaemia prevalence among pregnant teenagers.**

**S1 Table** shows the relationship between sociodemographic characteristics and anaemia prevalence. Anaemia prevalence did not significantly vary by sociodemographic factors (p>0.05). However, more rural (60.9%) than urban (54.4%) pregnant teenagers were anaemic (p = 0.222), older pregnant teenagers (16–19 years) were more anaemic than their younger (13–15 years) counterparts (57.6% versus 51.6%, p = 0.573), and the unmarried (59.4%) were more anaemic than the married participants (50.0%, p = 0.103).

Table 1 presents the relationship between iron deficiency status, and household hunger scale, lived poverty, intervention received at ANC, and micronutrient intakes. Over 8 in 10 pregnant teenagers (82.4%) with depleted body iron stores had inadequate women's dietary diversity score (WDDS) compared with less than 6 in 10 among those with normal body iron stores (57.4%, p = 0.046). Also, severe hunger was suffered by more pregnant teenagers with depleted iron store (29.2%), than those with normal body iron stores (11%, p = 0.029).

The proportion of participants who had depleted body iron stores and those with normal body iron stores did not significantly vary by nutrients intake (p>0.05), except for multiple micronutrient intake (iron-folate-vitamin B$_{12}$), whereby inadequate multiple micronutrient intake was higher among pregnant teenagers with depleted of body iron stores (82.4%) than those with normal body iron stores (57.4%, p = 0.026).

Table 2 indicates the relationship between MUAC status, serum nutrients status, and iron deficiency anaemia. The mean MUAC was higher among pregnant teenagers who were not anaemic (26.6±0.2cm) than those who were anaemic (25.8±0.2cm, p = 0.006). The proportion of participants who were anaemic and non-anaemic did not significantly differ by MUAC

**Table 1. Iron deficiency status, and household hunger scale, lived poverty index, intervention received at ANC, and micronutrient intakes.**

| Variables | Total | Hb status | | $X^2$, (P value) | Body iron stores | | $X^2$, (P value) |
|---|---|---|---|---|---|---|---|
| | | Anaemia | No anaemia | | Depleted | Normal | |
| **HHS[↓]** | | | | | | | |
| No Hunger | 303 (72.8) | 164(70.7) | 134(77.0) | 2.34(0.310)[¥] | 8(47.1) | 276(74.2) | 7.10(**0.029**)[‡] |
| Moderate Hunger | 63 (15.1) | 37(15.9) | 24(13.8) | | 4(23.5) | 55(14.8) | |
| Severe Hunger | 50 (12.0) | 31(13.4) | 16(9.2) | | 5(29.4) | 41(11.0) | |
| **Median(IQR) HHS[‡]** | 0(2.0) | 0(2.0) | 0(1.0) | 0.360 | 2(4.0) | 0(2.0) | 0.192 |
| **Lived Poverty Index(LPI)[↓]** | | | | | | | |
| Low Moderate | 194 (45.2) | 107(46.1) | 82(47.1) | 2.29(0.318)[¥] | 6(35.3) | 174(46.8) | 0.91(0.633)[‡] |
| High Moderate | 75 (18.3) | 38(16.4) | 37(21.3) | | 4(23.5) | 66(17.7) | |
| High | 152 (36.5) | 87(37.5) | 55(31.6) | | 7(41.2) | 132(35.5) | |
| **Median(IQR) LPI[‡]** | 1.2(0.8) | 1.4(1.0) | 1(0.6) | 0.118 | 1.6(1.4) | 1.2(0.8) | 0.597 |
| **WDDS[↓]** | | | | | | | |
| Inadequate | 247 (59.4) | 142(61.2) | 98(56.3) | 0.98(0.359)[‡] | 14(82.4) | 214(57.5) | 4.13(**0.046**)[‡] |
| Adequate | 169 (40.6) | 90(38.8) | 76(43.7) | | 3(17.6) | 158(42.5) | |
| **Median(IQR) WDDS[‡]** | 4(2.0) | 4(2.0) | 4(1.0) | 0.943 | 4(1.0) | 4(2.0) | 0.075 |
| **Take MS[↓]** | | | | | | | |
| Yes | 216(51.9) | 118(50.9) | 93(53.4) | 0.26(0.617)[‡] | 10(58.8) | 191(51.3) | 0.36(0.625)[‡] |
| No | 200(48.1) | 114(49.1) | 81(46.6) | | 7(41.2) | 181(48.7) | |
| **Take Malaria Tablet[↓]** | | | | | | | |
| Yes | 168(40.4) | 97(41.8) | 68(39.1) | 0.30(0.610)[‡] | 9(52.9) | 147(39.5) | 1.22(0.315)[‡] |
| No | 248(59.6) | 135(58.2) | 106(60.9) | | 8(47.1) | 225(60.5) | |
| **Dietary Micronutrients** | | | | | | | |
| **Dietary Iron, 23/22mg/d** | | | | | | | |
| Inadequate | 393(94.5) | 220(94.8) | 163(93.7) | 0.24(0.668)[‡] | 17(100.0) | 350(94.1) | 1.06(0.612)[‡] |
| Adequate | 23(5.5) | 12(5.2) | 11(6.3) | | 0(0.0) | 22(5.9) | |
| **Dietary Folate, 520μg/d** | | | | | | | |
| Inadequate | 345(82.9) | 194(83.6) | 143(82.2) | 0.14(0.790)[‡] | 16(94.1) | 307(82.5) | 1.55(0.326)[‡] |
| Adequate | 71(17.1) | 38(16.4) | 31(17.8) | | 1(5.9) | 65(17.5) | |
| **Dietary Vitamin B$_{12}$, 2.2μg/d** | | | | | | | |
| Inadequate | 226(54.3) | 123(53.0) | 97(55.7) | 0.29(0.615)[‡] | 13(76.5) | 196(52.7) | 3.69(00.080)[‡] |
| Adequate | 190(45.7) | 109(47.0) | 77(44.3) | | 4(23.5) | 176(47.3) | |
| **Dietary Iron-Folate** | | | | | | | |
| Inadequate | 343(82.5) | 193(83.2) | 142(81.6) | 0.17(0.694)[‡] | 16(94.1) | 305(82.0) | 1.65(0.327)[‡] |
| Adequate | 73(17.5) | 39(16.8) | 32(18.4) | | 1(5.9) | 67(18.0) | |
| **Dietary Iron-Vitamin B$_{12}$** | | | | | | | |
| Inadequate | 226(54.3) | 123(53.0) | 97(55.7) | 0.29(0.615)[‡] | 13(76.5) | 196(52.7) | 3.69(0.080)[‡] |
| Adequate | 190(45.7) | 109(47.0) | 77(44.3) | | 4(23.5) | 176(47.3) | |
| **MMI** | | | | | | | |
| Inadequate | 206(49.5) | 110(47.4) | 92(52.9) | 1.18(0.316)[‡] | 13(76.5) | 179(48.1) | 5.22(**0.026**)[‡] |
| Adequate | 210(50.5) | 122(52.6) | 82(47.1) | | 4(23.5) | 193(51.9) | |

↓Data are presented as frequency (percentage), ‡- Median(Interquartile), ‡- Mann Whitney test performed, ¥- Chi-square P value, ‡- Fisher's exact P value, Hb-Haemoglobin, HHS- Household Hunger Scale, WDDS- Women's Dietary Diversity Score, MS- Micronutrient Supplement, NE- Nutrition Education, MMI- Multiple Micronutrient Intakes, bold values are significant at p<0.05. EAR Sources: Dietary Reference Intakes for Thiamin, Riboflavin, Niacin, Vitamin B$_6$, Folate, Vitamin B$_{12}$, Pantothenic Acid, Biotin, and Choline [57]; Dietary Reference Intakes for Vitamin A, Vitamin K, Arsenic, Boron, Chromium, Copper, Iodine, Iron, Manganese, Molybdenum, Nickel, Silicon, Vanadium, and Zinc [58]. The EAR for iron for 19 years old: 23mg/day.

**Table 2. Relationship between MUAC status, biochemical parameters and iron deficiency anaemia status.**

| Variable | Total | Hb status | | $x^2$,P value | Body iron stores | | $x^2$, P value |
|---|---|---|---|---|---|---|---|
| | | Anaemia | No anaemia | | Depleted | Not depleted | |
| **Anthropometric** | | | | | | | |
| **MUAC status** | | | | | | | |
| **Mean MUAC** | 26.2±0.1 | 25.8±0.2 | 26.6±0.2 | **0.006**[‡] | 26.1±0.9 | 26.2±0.2 | 0.944[‡] |
| Severe wasting | 23(5.5) | 15(6.5) | 8(4.6) | 2.28 (0.319)[¥] | 1(5.9) | 22(5.9) | 1.58 (0.454)[¥] |
| Moderate/mild wasting | 97(23.3) | 58(25.0) | 35(20.1) | | 6(35.3) | 83(22.3) | |
| Normal | 296(71.2) | 159(68.5) | 131(75.3) | | 10(58.8) | 267(71.8) | |
| **Biochemical variable** | | | | | | | |
| **Serum vitamin A** | | | | | | | |
| **Median(IQR) Serum vit A** | 0.5(0.2) | 0.4(0.2) | 0.5(0.2) | 0.176[‡] | 0.4(0.7) | 0.5(0.2) | 0.837[‡] |
| Low | 336(86.4) | 194(87.8) | 142(84.5) | 0.86 (0.373)[‡] | 12(70.6) | 324(87.1) | 3.76 (0.066)[‡] |
| Normal | 53(13.6) | 27(12.2) | 26(15.5) | | 5(29.4) | 48(12.9) | |
| **Serum ferritin status** | | | | | | | |
| **Mean serum ferritin** | 36.0±0.8 | 36.7±1.2 | 35.1±1.2 | 0.305[‡] | 17.4±0.7 | 36.9±0.8 | **<0.001**[‡] |
| Low | 14(3.6) | 9(4.1) | 5(3.0) | 0.33 (0.785) [‡] | 1(5.9) | 13(3.5) | 0.26 (0.471)[‡] |
| Normal | 375(96.4) | 212(95.9) | 163(97.0) | | 16(94.1) | 359(96.5) | |
| **Serum PAB** | | | | | | | |
| **Mean serum PAB** | 12.4±0.7 | 12.3±0.9 | 12.6±1.0 | 0.777[‡] | 8.0±0.5 | 12.7±0.7 | **<0.001**[‡] |
| Low | 382(97.2) | 216(96.9) | 165(97.6) | 0.21 (0.763) [‡] | 17(100.0) | 361(97.0) | 0.51 (1.000)[‡] |
| Normal | 11(2.8) | 7(3.1) | 4(2.4) | | 0(0.0) | 11(3.0) | |
| **ZPP/Hb ratio** | | | | | | | |
| **Median(IQR) ZPP/Hb ratio** | 67.2(32.2) | 79.2(29.4) | 53.8(21.1) | **<0.001**[‡] | 96.4(21.0) | 66.5(32.0) | **<0.001**[‡] |
| Normal | 150(38.6) | 37(16.7) | 113(67.3) | 118.33(**<0.001**)[¥] | 1(5.9) | 149(40.1) | 32.59 (**<0.001**) [¥] |
| Moderately high | 117(30.1) | 75(33.9) | 42(25.0) | | 0(0.0) | 117(31.5) | |
| High | 122(31.4) | 109(49.3) | 13(7.7) | | 16(94.1) | 106(28.5) | |
| **CRP status** | | | | | | | |
| Normal | 371(95.4) | 211(95.5) | 160(95.2) | 0.01(1.000)[‡] | 16(94.1) | 355(95.4) | 0.063(0.561)[‡] |
| High | 18(4.6) | 10(4.5) | 8(4.8) | | 1(5.9) | 17(4.6) | |
| **Median(IQR) serum CRP** | 3.2(1.3) | 3.1(1.4) | 3.2(1.2) | 0.242[‡] | 3.2(1.8) | 3.2(1.3) | 0.991[‡] |
| **Median(IQR) ZPP** | 26.2(9.6) | 28.6(9.4) | 23.3(8.4) | **<0.001**[‡] | 32.7(8.7) | 25.9(9.4) | **0.046**[‡] |
| **Median(IQR) serum TAC** | 1.1(0.6) | 1.1(0.6) | 1.1(0.6) | 0.821[‡] | 1.0(0.8) | 1.1(0.6) | 0.822[‡] |

[┼]Data are presented as frequency (percentage), ‡- Mean±SEM (standard error mean), [‡]Independent t-test, [‡]Median (IQR- Interquartile) [‡]- Mann Whitney test, [¥]- Chi-square P value, [‡]- Fisher's exact P value, Hb- Haemoglobin, MUAC- Mid-Upper Arm Circumference, PAB- Prealbumin, ZPP- Zinc protoporphyrin, CRP- C-reactive protein, TAC- Total antioxidant capacity, bold values are significant at p < 0.05. ZPP/Hb measured in µmol/mol heme.

status (p = 0.319), although severe wasting was higher among anaemic participants (6.5%) than non-anaemic (4.6%) participants.

The proportion of participants with a high ZPP/Hb ratio, indicating poor systemic iron supply, was higher in anaemic (49.3%), compared with non-anaemic (7.7%) participants (p<0.001). The median levels of serum ZPP (p<0.001), and ZPP/Hb ratio (p<0.001) were significantly higher among pregnant teenagers who were anaemic (28.6µmol/L, 79.2µmol/mol heme) than those who were not anaemic (23.3µmol/L, 53.8 µmol/mol heme) respectively. For serum iron status, more anaemic participants (6.8%) had depleted iron stores than those who were not anaemic (1.2%, p = 0.010).

Almost all participants with deficient systemic iron supply (high ZPP/Hb ratio) (94.1%) had depleted body iron stores compared with less than a third (28.5%) among those with

normal body iron stores (p<0.001). The mean levels of serum ferritin (36.9±0.8μg/L versus 17.4±0.7μg/L p<0.001), and prealbumin (12.7±0.7mg/L vs 8.0±0.5mg/L, p = 0.007) were higher among pregnant teenagers with normal body iron stores than those who had depleted body iron stores, respectively. Moreover, participants with normal body iron stores had a lower median level of serum ZPP (32.7μmol/L vs 25.9μmol/L, p = 0.046), and ZPP/Hb ratio (96.4μmol/mol heme vs 66.5μmol/mol heme, p<0.001), compared to those who had depleted body iron stores respectively.

Table 3 indicates how dietary factors and hunger status are associated with nutritional status and anaemia. Among those with inadequate MMI and depleted body iron stores, 84.6% were anaemic compared with 54.5% anaemia in adequate MMI and normal body iron stores. Among adequate MMI but depleted body iron stores, anaemia was 100% compared with 56.2% anaemia among inadequate MMI and normal body iron store participants. Anaemia in no hunger iron-depleted participants was 100%, moderate hunger iron-depleted stores participants 50%, and severe hunger iron-depleted stores participants 100%, compared to lower levels in no hunger normal iron stores (53.6%), moderate hunger normal body iron stores (60%), and severe hunger normal body iron stores participants (61%), p = 0.010. Likewise, anaemia was higher in inadequate WDDS and also depleted body iron stores participants (85.7%), than inadequate WDDS but normal body iron stores participants (56.4%), and also higher in adequate WDDS but depleted body iron stores participants (100%), compared with normal body iron stores (53.8%), p = 0.010.

Moreover, anaemia was consistently highest among high ZPP/Hb ratio participants (high ZPP/Hb signifying poor systemic iron supply), with either depleted (93.8%) or normal iron stores (88.7%), compared with the moderate ZPP/Hb ratio group, and the normal ZPP/Hb ratio group.

Last but not the least, anaemia was highest among low serum ferritin-depleted body iron stores participants (100%), followed by normal serum ferritin-depleted body iron stores (87.5%), then low serum ferritin-normal body iron stores (61.5%), and the least were among normal ferritin with normal body iron stores (55.2%, p = 0.010).

The results of between-subject effects, conducted in the univariate Generalized Linear Model are presented in Table 4. Taking into account all the multiple underlying factors of anaemia in the tests of associations on Hb levels, it was malaria tablet given at antenatal clinic (p = 0.035), MUAC (p = 0.043), ZPP (p<0.001), ZPP/Hb ratio (p<0.001) and body iron stores (p<0.001) that had a significant effect on the Hb levels. Some variables (e.gs. lived poverty index, serum levels of vitamin A, C-reactive protein, prealbumin) which were not significant were removed, to report a few items in the table.

The predictors of anaemia and depleted body iron stores of pregnant teenagers are presented in Table 5. Pregnant teenagers who had a high ZPP/Hb ratio (signifying poor systemic iron supply) had more than 9 times increased odds of being anaemic (OR = 9.7, p<0.001, 95% CI = 6.0–15.8), compared to those who had normal ZPP/Hb ratio. Pregnant teenagers who were wasted (OR = 1.2, p = 0.543, 95%CI = 0.6–2.3), and those who had depleted iron stores (OR = 3.0, p = 0.167, 95%CI = 0.6–14.6), had increased odds of being anaemic, compared to those with normal MUAC or normal iron stores respectively.

Also, pregnant teenagers who experienced hunger had over 2 times increased odds of having depleted body iron stores (OR = 2.9, p = 0.040, 95%CI = 1.1–7.8), compared to those who did not experience hunger. Participants who had inadequate multiple micronutrients intakes (OR = 2.6, p = 0.102, 95%CI = 0.8–8.4), and those with low serum levels of ferritin (OR = 3.3, p = 0.291, 95%CI = 0.4–29.2), had an increased odds of depleted body iron stores. Pregnant teenagers who had inadequate women's dietary diversity score (OR = 0.3, p = 0.087, 95% CI = 0.1–1.2) had a lower odds of depleted body iron stores.

**Table 3. Interplay between MMI, HHS, WDDS, ZPP/Hb ratio, serum ferritin, body iron stores, and Hb status.**

| | | Haemoglobin (Hb) status | | | |
|---|---|---|---|---|---|
| **MMI** | **Variable** | **Anaemia** | **No anaemia** | **Chi-square** | **P value** |
| Inadequate | **Body Iron stores** | | | | |
| | Depleted | 11(84.6%) | 2(13.4%) | 7.154 | **0.010** |
| | Normal | 102(54.5%) | 85(45.5%) | | |
| Adequate | **Body Iron stores** | | | | |
| | Depleted | 4(100) | 0(0.0) | | |
| | Normal | 104(56.2) | 81(53.8) | | |
| **HHS** | **Body Iron stores** | | | | |
| No hunger | Depleted | 8(100) | 0(0.0) | 7.154 | **0.010** |
| | Normal | 148(53.6) | 128(46.4) | | |
| Moderate | **Body Iron stores** | | | | |
| | Depleted | 2(50) | 2(50) | | |
| | Normal | 33(60) | 22(40) | | |
| Severe | **Body Iron stores** | | | | |
| | Depleted | 5(100) | 0(0.0) | | |
| | Normal | 25(61) | 16(39) | | |
| **WDDS** | **Body Iron stores** | | | | |
| Inadequate | Depleted | 12(85.7) | 2(14.3) | 7.154 | **0.010** |
| | Normal | 121(56.4) | 93(43.6) | | |
| Adequate | **Body Iron stores** | | | | |
| | Depleted | 3(100) | 0(0.0) | | |
| | Normal | 85(53.8) | 73(46.2) | | |
| **ZPP/Hb ratio** | **Body Iron stores** | | | 7.154 | **0.010** |
| Normal | Depleted | 0(0.0) | 1(100) | | |
| | Normal | 37(24.8) | 112(75.2) | | |
| Moderate high | **Body Iron stores** | | | | |
| | Depleted | 0(0.0) | 0(0.0) | | |
| | Normal | 75(64.1) | 42(35.9) | | |
| High | **Body Iron stores** | | | | |
| | Depleted | 15(93.8) | 1(6.2) | | |
| | Normal | 94(88.7) | 12(11.3) | | |
| **Serum ferritin** | **Body Iron stores** | | | | |
| Low | Depleted | 1(100) | 0(0.0) | 7.154 | **0.010** |
| | Normal | 8(61.5) | 5(38.5) | | |
| Normal | **Body Iron stores** | | | | |
| | Depleted | 14(87.5) | 2(11.5) | | |
| | Normal | 198(55.2) | 161(44.8) | | |

Data are presented as frequency (percentage), Fisher's exact P value, Hb- Haemoglobin, MMI- Multiple Micronutrient Intake, HHS- Household Hunger Scale, WDDS-Women's Dietary Diversity Score, bold values are significant at p < 0.05.

## Discussion

The key findings of the study were the high prevalence of iron deficiency anaemia among teenagers. Iron deficiency anaemia (IDA) was common among pregnant teenagers, from both systemic serum iron, and their diet, and this is worrying. Our finding on IDA among teenagers is higher than that of pregnant adults in other studies in Ghana [20,59,60]. The studies by the Ghana Micronutrients Survey [20], the Ghana Demographic Health Survey [59], and Ayensu

**Table 4. Tests of between-subject effects of multiple variables on haemoglobin levels.**

| Tests of Between-Subjects Effects | | | | | |
|---|---|---|---|---|---|
| **Dependent Variable: Hb levels** | | | | | |
| **Source** | **Type III Sum of Squares** | **Df** | **Mean Square** | **F** | **Significance** |
| Intercept | 47.54 | 1 | 47.54 | 44.645 | <0.001 |
| Community type | 0.886 | 1 | 0.886 | 0.832 | 0.362 |
| Age | 0.559 | 1 | 0.559 | 0.525 | 0.469 |
| WDDS | 0.008 | 1 | 0.008 | 0.007 | 0.933 |
| HHS | 0.032 | 1 | 0.032 | 0.03 | 0.862 |
| Malaria tablet given | 4.788 | 1 | 4.788 | 4.497 | **0.035** |
| Dietary iron | 0.758 | 1 | 0.758 | 0.712 | 0.399 |
| MMI | 0.05 | 1 | 0.05 | 0.047 | 0.829 |
| MUAC | 4.392 | 1 | 4.392 | 4.125 | **0.043** |
| Serum ferritin | 0.422 | 1 | 0.422 | 0.396 | 0.529 |
| ZPP | 158.095 | 1 | 158.095 | 148.469 | <**0.001** |
| ZPP/Hb ratio | 188.605 | 1 | 188.605 | 177.122 | <**0.001** |
| Body iron store | 18.086 | 1 | 18.086 | 16.985 | <**0.001** |
| Error | 387.599 | 364 | 1.065 | | |

General Linear Model test for univariate parameters, BMI- Body mass index, Hb- Haemoglobin, TMS- Took Micronutrient Supplement, CMI- Combined Macronutrient Intake (energy, carbohydrate, and protein), MMI- Multiple Micronutrient Intake (Iron, folate and vitamin $B_{12}$), ZPP- Zinc protoporphyrin, Bold values are significant at p<0.05.

et al. [60] have reported that anaemia prevalence among pregnant adults were 56.0%, 44.6%, and 42.0% respectively. In Ghana, Ampiah et al. [21] and Nonterah et al. [32] reported that 70% and 52.0% pregnant teenagers were anaemic respectively, while in Palestine, a study by Srour et al. [61] reported 25.0% pregnant teenagers were anaemic and 52.0% had depleted body iron stores. This means that compared to adults, teenagers are more likely to be anaemic during pregnancy from our findings, and this may be due to their high nutritional needs and immature body tissue growth. Anaemia in pregnant teenagers has been attributed to the fact that they are still growing and need additional iron and folic acid to meet their own nutritional needs, and those of the developing foetus during gravidity [16]. However, we also report very low intakes of nutrients such as iron, folate and vitamin $B_{12}$, and all these can lead to anaemia. Hence the high prevalence of anaemia in this group could be due to low dietary intake of nutrients/poor diet diversity that prevent anaemia. Iron deficiency anaemia is a public health problem among pregnant teenagers in Ghana [32]. Anaemia in pregnancy can lead to a greater risk of poor pregnancy outcomes, such as low birth weight, small-for-gestation age, stillbirth, preterm birth [16]. Our findings therefore suggest that the teenagers are at a greater risk of adverse birth outcomes.

Food deprivation can increase the risk of low systemic iron stores, and consequently anaemia [25]. For the adolescent, pregnancy can be a complicated issue due to the rapidly increased demands of nutrients, especially for iron and folic acid, for the mother's blood expansion, growth as well as the foetus. Food deprivation and poor access to food are linked with poor dietary intakes, and the risk of micronutrient deficiencies, which can consequently lead to depleted body stores of nutrients [24,27,62]. We found hunger to predict depleted iron stores, with teenagers who suffered hunger, compared to those who did not suffer from hunger at increased risk of depleted body iron stores (OR = 2.9, p = 0.040, 95%CI = 1.1–7.8). Our previous study showed that sever hunger of the pregnant teenagers were associated with inadequate dietary diversity; and poor diet diversity could have contributed to their depleted iron stores

**Table 5.  Predictors of anaemia and depleted body iron stores.**

| | | Anaemia | | 95% Confidence Interval | |
|---|---|---|---|---|---|
| Variable | β | OR | Significance | Lower | Upper |
| **MUAC** | | | | | |
| Wasted | 0.201 | 1.2 | 0.543 | 0.6 | 2.3 |
| Normal | | 1.0 | | | |
| **ZPP/Hb ratio** | | | | | |
| High | 2.277 | 9.7 | <**0.001** | 6.0 | 15.8 |
| Normal | | 1.0 | | | |
| **Body iron stores** | | | | | |
| Depleted | 1.109 | 3.0 | 0.167 | 0.6 | 14.6 |
| Normal | | 1.0 | | | |
| | | Depleted body iron stores | | 95% Confidence Interval | |
| Variable | β | OR | Significance | Lower | Upper |
| **Serum ferritin** | | | | | |
| Low | 1.181 | 3.3 | 0.291 | 0.4 | 29.2 |
| Normal | | 1.0 | | | |
| **Serum Prealbumin** | | | | | |
| Low | -17.346 | 0.01 | 0.999 | | |
| Normal | | 1.0 | | | |
| **MMI** | | | | | |
| Inadequate | 0.967 | 2.6 | 0.102 | 0.8 | 8.4 |
| Adequate | | 1.0 | | | |
| **HHS** | | | | | |
| Hunger present | 1.049 | 2.9 | **0.040** | 1.1 | 7.8 |
| No hunger | | 1.0 | | | |
| **WDDS** | | | | | |
| Inadequate | -1.136 | 0.3 | 0.087 | 0.1 | 1.2 |
| Adequate | | 1.0 | | | |
| ZPP | -0.004 | 1.0 | 0.493 | 0.9 | 1.0 |

**β** –Slope/regression co-efficient, OR- Odds ratio, Hb- Haemoglobin, combined moderate and severe wasting, moderate and severe hunger, and moderate-high and high ZPP/Hb ratio for statistical strength bold values are significant at p < 0.05.

[63]. We also found participants who were anaemic were more likely to suffer from both severe hunger and depleted body iron stores, suggesting a combined effect. This implies that severe maternal hunger could have caused depleted body iron stores, increasing the risk for anaemia. Although we observed that malaria tablets received at ANC effectively determined haemoglobin levels of the participant, those who were given tended to be more anaemic, implying that malaria was not the likely cause of iron deficiency anaemia among the teenagers.

The body requires iron, vitamin $B_{12}$, and folate as substrates and co-factors for the effective production of haemoglobin. Meanwhile, poor dietary quality or low dietary diversity are the main culprit for low vitamin and mineral levels including vitamin $B_{12}$, folate, and iron, increasing risk for anaemia [25,64]. We observed that teenagers who had inadequate WDDS compared to those with adequate WDDS were more likely to have a depleted body iron store (p = 0.046). Inadequate WDDS among the teenagers significantly contributed to poorer iron stores and anaemia occurrence, although the regression model showed a lower risk for depleted body iron stores (OR = 0.3, p = 0.089, 95%CI = 0.1–1.2). We also found that pregnant teenagers with depleted body iron stores were more likely to have inadequate multiple

micronutrient intake than those with normal body iron stores (p = 0.046), and that, inadequate multiple micronutrient intakes (iron-folate-vitamin B$_{12}$) contributed to depleted body iron stores and anaemia occurrence in the teenagers. This implies that consumption of less diversified diet, which could manifest as inadequate multiple micronutrient intake might have led to depleted iron stores, leading anaemia. In Indonesia, Diana et al. [65] reported the consumption of a less diversified diet, which lacked dietary iron, contributed to anaemia in pregnant women. Our findings call for strengthening nutrition education at the antenatal care clinics to promote the consumption of nutritious and diversified diets, but also improving the iron-folate supplementation offered to pregnant women.

Most often, women in low-and-middle-income countries go into pregnancy malnourished, and the demands of gravidity can worsen micronutrient deficiencies, leading to iron deficiency anaemia [32]. In this study maternal MUAC was associated with iron deficiency anaemia. The participants who were wasted (using MUAC) (OR = 1.2, p = 0.543, 95%CI = 0.6–2.3) were more likely to be anaemic, and this was consistent with a study among pregnant adults in Ghana [26]. This implies that wasting can be associated with anaemia during pregnancy.

We observed that being depleted of body iron stores compared to normal body iron stores was associated with an increased risk of anaemia. Deficient systemic supply of iron, and depleted body iron stores, were both predictors of iron deficiency anaemia in the teenagers, with an odds ratio greater than 2. Participants with a deficient supply of iron were more likely to have poorer iron stores and become anaemic. Similarly, participants with low serum ferritin levels were more likely to have poorer iron stores and become anaemic. These imply that low serum ferritin and deficient systemic supply of iron could also be associated with depleted iron stores in the teenagers, thus lead to the occurrence of maternal anaemia. Poor dietary intake might contribute to the depletion of iron stores, which affects the systemic supply of iron for haemoglobin production [25,64]. In the body, ZPP and serum ferritin are among iron biomarkers that depend on dietary intake to affect the iron status, and the combination of the two can predict whether iron stores are depleted or not [56]. In cases where iron is deficient from body stores (that is; low serum ferritin), the final step of haemoglobin production which involves a combination of ferrous protoporphyrin and globin is affected, leading to iron being replaced by zinc to produce zinc protoporphyrin (ZPP) in a normal ratio 30,000:1. This progressive iron deficiency increases ZPP levels, which has a profound effect on the supply of iron for haemoglobin production [47,66]. In this study, we observed a high ZPP/Hb ratio among the teenagers who were anaemic, indicating deficient iron supply for haemoglobin production. All these associations tend to explain that poor micronutrient intake and severe hunger, and low serum ferritin could contribute to deficient systemic iron supply and concurrent depleted body iron stores, leading to iron deficiency anaemia among pregnant teenagers.

Our findings have implications. In our previous study, which involved a secondary data analysis of anaemia trend in the district, we reported a higher prevalence of anaemia among pregnant teenagers than adults, and mean Hb levels of both teenagers and adults were found to be higher by the 36th week of pregnancy than at antenatal registration. We observed a positive correlation between Hb at antenatal registration and 36th-week gestation, indicating that pre-pregnancy Hb levels, largely determined anaemia occurrence during the latter stages of pregnancy [21]. In this follow up study, we sought to understand the associated factors of anaemia among teenagers the teenagers. This group is understudied, and this makes our study and its findings crucial. The current findings have provided enough evidence that the high maternal anaemia prevalence among teenagers was a result of hunger experienced during pregnancy, inadequate multiple micronutrient intakes related to poor dietary quality or low dietary diversity, and low serum ferritin levels, which contributed to deficient systemic supply of iron and depleted body iron stores. Other factors, such as wasting and inadequate dietary diversity

contributed to depleted iron stores, and anaemia occurrence. Malaria tablet given at ANC was not significantly associated with Hb levels. Our findings call for the need for nutrition education, livelihood empowerment among these teenager, nutritional support to meet their needs, and strengthening antenatal service delivery to pregnant adolescents. Also, interventions that encourage food fortification and diversification has been reported to reduce anaemia in women of child-bearing age, and these interventions can be considered [67].

## Study limitations

The study has limitations. The use of 24 h dietary recalls may lead to over-or under-estimations [68]. In the present study, we used a 3-day repeated recall to cover up for this. Also, a couple of food items consumed were not among the listed foods in the nutrient analysis template used to estimate nutrient levels. Instead, the template had similar foods that were used. This could have affected the estimated nutrient levels. Interpretation of the findings may not be generalized to the rest of Ghana, but probably indicate a similar situation of adolescents in other Regions of Ghana. We are aware of some bias in using a first come first served procedure in sampling. However, we employed convenience sampling by using a first come first served basis due to the low turnout of the pregnant teenagers at the clinics. As such, this sampling method was therefore appropriate to help obtain the study sample size and a representative population. Except for participants who were ineligible for this study, all participants who came and were eligible were recruited. Thus, there is less likely to have a probability bias risk involved in the sampling process.

## Conclusions

Iron deficiency anaemia is common among the pregnant teenagers studied, and the related significantly to hunger, low MUAC, a deficient systemic supply of iron, depleted body iron stores, and to a smaller extent malaria tablets given at ANC. Antenatal service delivery should focus on the nutritional needs of pregnant women, while special attention is given to teenagers to prevent the devastating consequences of anaemia during pregnancy.

## Supporting information

**S1 Table. Relationship between sociodemographic and haemoglobin status.**
(DOCX)

## Acknowledgments

We are grateful for the support received from the directors of health, health workers in the health centers where the study took place, and the pregnant teenagers.

## Author Contributions

**Conceptualization:** Charles Apprey, Anthony Kwaku Edusei, Wisdom Azanu, Herman E. Lutterodt.

**Data curation:** Reginald Adjetey Annan, Linda Afriyie Gyimah, Charles Apprey, Odeafo Asamoah-Boakye, Linda Nana Esi Aduku.

**Formal analysis:** Reginald Adjetey Annan, Linda Afriyie Gyimah, Odeafo Asamoah-Boakye.

**Funding acquisition:** Reginald Adjetey Annan, Charles Apprey.

**Investigation:** Reginald Adjetey Annan, Linda Afriyie Gyimah, Charles Apprey, Anthony Kwaku Edusei, Odeafo Asamoah-Boakye, Linda Nana Esi Aduku, Wisdom Azanu, Herman E. Lutterodt.

**Methodology:** Reginald Adjetey Annan, Linda Afriyie Gyimah, Charles Apprey, Anthony Kwaku Edusei, Linda Nana Esi Aduku, Wisdom Azanu, Herman E. Lutterodt.

**Project administration:** Reginald Adjetey Annan, Linda Nana Esi Aduku.

**Resources:** Reginald Adjetey Annan, Linda Afriyie Gyimah, Charles Apprey, Linda Nana Esi Aduku, Wisdom Azanu, Herman E. Lutterodt.

**Software:** Reginald Adjetey Annan, Odeafo Asamoah-Boakye.

**Supervision:** Reginald Adjetey Annan, Charles Apprey, Anthony Kwaku Edusei, Linda Nana Esi Aduku, Herman E. Lutterodt.

**Validation:** Linda Afriyie Gyimah, Charles Apprey, Wisdom Azanu.

**Visualization:** Reginald Adjetey Annan, Linda Afriyie Gyimah, Charles Apprey, Wisdom Azanu, Herman E. Lutterodt.

**Writing – original draft:** Reginald Adjetey Annan, Odeafo Asamoah-Boakye.

**Writing – review & editing:** Reginald Adjetey Annan, Linda Afriyie Gyimah, Charles Apprey, Anthony Kwaku Edusei, Linda Nana Esi Aduku, Wisdom Azanu, Herman E. Lutterodt.

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
