## [Decision Letter · Decision Letter 0]

11 Sep 2020

PONE-D-20-21782

Factors associated with Iron Deficiency Anaemia among pregnant teenagers in Ghana

PLOS ONE

Dear Dr. Annan,

Thank you for submitting your manuscript to PLOS ONE. After careful consideration, we feel that it has merit but does not fully meet PLOS ONE’s publication criteria as it currently stands. Therefore, we invite you to submit a revised version of the manuscript that addresses the points raised during the review process.

No post-hoc corrections for multiple comparisons are referred to in this manuscript.

The potential limitations of the study are not discussed.

We look forward to receiving your revised manuscript.

Kind regards,

Dr Nitin Joseph

Academic Editor

PLOS ONE

Journal Requirements:

2. Please address the following:

- Please refer to any post-hoc corrections to correct for multiple comparisons during your statistical analyses. If these were not performed please justify the reasons. Please refer to our statistical reporting guidelines for assistance (https://journals.plos.org/plosone/s/submission-guidelines.#loc-statistical-reporting).

- Please ensure you have thoroughly discussed any potential limitations of this study within the Discussion section, including the potential introduction of bias during data collection.

3. You indicated that you had ethical approval for your study and state: "written personal or parental consent was obtained from participants before the study". Please confirm that you obtained consent from parents or guardians of the minors included in the study specifically.

Reviewers' comments:

Reviewer's Responses to Questions

**Comments to the Author**

1. Is the manuscript technically sound, and do the data support the conclusions?

Reviewer #1: Yes

Reviewer #2: Yes

Reviewer #3: Yes

Reviewer #4: Partly

2. Has the statistical analysis been performed appropriately and rigorously? 

Reviewer #1: No

Reviewer #2: Yes

Reviewer #3: Yes

Reviewer #4: Yes

3. Have the authors made all data underlying the findings in their manuscript fully available?

Reviewer #1: Yes

Reviewer #2: No

Reviewer #3: Yes

Reviewer #4: Yes

4. Is the manuscript presented in an intelligible fashion and written in standard English?

Reviewer #1: Yes

Reviewer #2: Yes

Reviewer #3: Yes

Reviewer #4: Yes

5. Review Comments to the Author

Reviewer #1: Title -Please add the type of study

Key words -Should be mesh term

Ethics section:If the teenage pregnant women is less than 18 then was consent of parents pr gaurdians taken??

In analysis Fishers exact test is used for 3x2 tables also but fishers exact test should be used for 2x2 table

Reviewer #2: Some methodological queries including validation of the data collection tools, the constitution of the team that performed the data collection, training provided to the team, standardization of data collection methods (including anthropological and biochemical investigations) have not been addressed.

Limitations of the study, potential sources of bias including those due to sampling and efforts made, if any, to address the same are not reported.

In the results section, there is repetition of information in the text and the tables, which can be avoided- only relevant findings and statistically significant associations may be included in the text.

The number of tables may be decreased (Comments attached within the pdf of the manuscript).

Discussion may be made more crisp and concise.

Reviewer #3: The answers to the comments have been included along with the questions.

If Non Parametric test has been used (Mann Whitney U test), then continuous variables may be expressed as Median and Inter quartile range.

For variables such as mean MUAC, mean Serum PAB, mean Serum Feeritin, it seems that Independent t test is appropriate rather than Mann Whitney U test as mentioned in Table 2 and Table3

Reviewer #4: IDA has been thoroughly researched and it has been found that most common cause is nutritional, it was not clear in the need as to why authors sought those particular factors for study. Study results were highlighted, however discussion with known literature could be better. Limitations of the study and generalizability and feasibility were not discussed well. Few sentences could be phrased better.

6. PLOS authors have the option to publish the peer review history of their article (what does this mean?). If published, this will include your full peer review and any attached files.

Reviewer #1: No

Reviewer #2: No

Reviewer #3: No

Reviewer #4: No

---

## [Author Response · Author response to Decision Letter 0]

9 Mar 2021

Dear Editor-in-Chief,

Thanks for reviewing our manuscript.

In line with the reviewer’s comments, the following revisions have been made to the Manuscript ID: PONE-D-20-21782 entitled “Factors associated with Iron Deficiency Anaemia among pregnant teenagers in Ghana: A hospital-based prospective cohort study’ by Reginald Adjetey Annan, Linda Afriyie Gyimah, Charles Apprey, Anthony Kwaku Edusei, Odeafo Asamoah-Boakye, Linda Nana Esi Aduku, Wisdom Azanu, and Herman E. Lutterodt. Below is our submission, highlight in blue colour, and yellow in the revised manuscript.

Journal Requirements/Editor comments:

Response: The manuscript has been revised to meet all of PLOS ONE’s style requirements. 

 2. Please address the following:

- Please refer to any post-hoc corrections to correct for multiple comparisons during your statistical analyses. If these were not performed please justify the reasons. Please refer to our statistical reporting guidelines for assistance (https://journals.plos.org/plosone/s/submission-guidelines.#loc-statistical-reporting).

Response: The manuscript had no multiple comparison that used One-way ANOVA, and so post-hoc corrections were not needed to correct multiple comparisons. 

- Please ensure you have thoroughly discussed any potential limitations of this study within the Discussion section, including the potential introduction of bias during data collection.

Response: The discussion has been revised to include limitations during data collection

3. You indicated that you had ethical approval for your study and state: "written personal or parental consent was obtained from participants before the study". Please confirm that you obtained consent from parents or guardians of the minors included in the study specifically.

Response: Yes, parents/guardians of the teenagers gave consent on behalf of participants less than 18 years old.

Response: Yes, there are no legal or ethical restriction to our data, and will be made available upon request..

Response: Not applicable 

Response: Data will be made available upon acceptance for publication.

Response to Reviewer One Comments

Point 1: Title -Please add the type of study

Response 1: The type of study has been added to the manuscript title. Please see lines 1-2

Point 2: Keywords -Should be mesh term

Response 2: Keywords has been revised to use mesh term. Please see line 46

Point 3: Ethics section: If the teenage pregnant women are less than 18 then was consent of parents/guardians taken??

Response 3: Yes, we obtained consent from parents/guardians of participants less than 18 years. Please we have included this statement in the Ethics section. Please see line 154-155.

Point 4: In analysis Fishers exact test is used for 3x2 tables also but fishers exact test should be used for 2x2 table.

Response 4: Yes, Fisher’s exact is required for 2 x 2 Cross-tabulation analysis, while Chi-square test is required for 3 x 2 onwards. However, in cases when the cell values are less than 5 in 3 x 2 Cross-tabulation analysis, the Chi-square test is violated and void and thus the Fisher’s exact P value is rather preferred, and reported.

Point 6: Mention the full form first time in text

Response 6: The full form of MUAC has been mentioned in the methods of Abstract, and abbreviation has been subsequently used. Please see line 23.

Point 7: Why was 32 weeks kept as the cut off?

Response 7: Initially, the cut-off for recruiting the pregnant teenagers was set at below 24 months but we were unable to get enough participants during the early stage of the data collection, and the study had a timeline for each stage of the research. Due to this situation encountered, the gestation age for recruitment was increased to 32 weeks to give more room for recruitment.

Point 8: Why was low birth weight selected for sample size calculation?

Response 8: This study is part of a larger prospective study that followed the teenagers to obtain birth outcomes. The sample size was calculated from the main study which had birth weight as the main outcome of the larger study. We used the prevalence rate of low birth weight from a previous study in Ghana in calculating the sample size. We have included this statement in the sample size determination to give a clearer picture. Please see lines 119-121.

Point 9: Not clear in eligibility criteria

Response 9: The statement on the eligibility criteria has been revised to make it clear. The inclusion criterion was that pregnant teenagers should be within the ages 13 – 19 years old, and also a resident in the selected communities. Please see lines 141-143.

Point 10: Figure attached is not clear, also axis is not appropriate.

Response 10: The figure has been revised to have a clearer view, and the percentage axis has been removed. Please see figure 1 at lines 742-745.

Point 11: How was Fisher’s test used in Supplement 1 for Income status and haemoglobin status?

Response 11. We used Fisher’s exact test because some cell values were less than 5 and in such case, the Chi-square test is violated. 

Response to Reviewer Two Comments

Reviewer #2 Point 1: Some methodological queries including validation of the data collection tools, the constitution of the team that performed the data collection, training provided to the team, standardization of data collection methods (including anthropological and biochemical investigations) have not been addressed.

Response 1. Please the methodological queries raised in the data collection section have been addressed. Please see lines 162-166; 169-170; 207-209; 214-215, 217-219, 226-227 under Data collection section. 

Point 2: Limitations of the study, potential sources of bias including those due to sampling and efforts made, if any, to address the same are not reported.

Response 3: Thanks for prompting our attention to this important issue. The limitation of the study has been reported in the manuscript. The study has limitations. The use of 24 h dietary recalls may lead to over-or under-estimations [Arsenault et al., 2020]. In the present study, we used a 3-day repeated recall to cover up for this. Also, a couple of food items consumed were not among the listed foods in the nutrient analysis template used to estimate nutrient levels. Instead, the template had similar foods that were used. This could have affected the estimated nutrient levels. Interpretation of the findings may not be generalized to the rest of Ghana, but probably indicate a similar situation of adolescents in other Regions of Ghana. We are aware of some bias in using a first come first served procedure in sampling. However, we employed convenience sampling by using a first come first served basis due to the low turnout of the pregnant teenagers at the clinics. As such, this sampling method was, therefore, appropriate to help obtain the study sample size and a representative population. Except for participants who were ineligible for this study, all participants who came and were eligible were recruited. Thus, there is less likely to have a probability bias risk involved in the sampling process. Please see lines 527-538.

Point 4: In the results section, there is repetition of information in the text and the tables, which can be avoided- only relevant findings and statistically significant associations may be included in the text. The number of tables may be decreased (Comments attached within the pdf of the manuscript).

Discussion may be made more crisp and concise.

Response 4: The description of the table results has been revised to describe only relevant and significant findings. Tables 1,2,3 and supplement 2 have been combined and restructured to reduce the number of tables. Please see new Tables 1-3. The discussion has been revised to make it concise.

Point 5: Risk of selection bias, how did authors address this bias

Response 5: We are aware of some bias in using a first come first served procedure in sampling. However, we employed convenience sampling by using a first come first served basis due to the low turnout of the pregnant teenagers at the clinics. As such, this sampling method was, therefore, appropriate to help obtain the study sample size and a representative population. Except for participants who were ineligible for this study, all participants who came and were eligible were recruited. Thus, there is less likely to have a probability bias risk involved in the sampling process. Please see lines 532-538.

Point 6: Who collected the data on anthropometric measurement? How many were involved? What training was received?

Response 6: Anthropometric measurements were taken by trained enumerators. A two-day training workshop was undertaken by research experts to train all enumerators on each data collection tool. A day field pre-testing session followed the training in a nearby community health center. These trained enumerators carried out all data collection for this study at the health centers/hospitals. Please see lines 161-166; 207-208.

Point 7: Who administered the questionnaire? How was the questionnaire validated? May not be the most appropriate method, could have gone for 1 week recall- dietary cycle method.

Response 7: Trained MPhil Human Nutrition and Dietetics students collected dietary intakes of the participants. Please see line 169-171. The 24-hour dietary recall is a validated tool, which has been extensively used in assessing dietary intakes of the population (Arsenault et al., 2020). We are also aware of the disadvantages in the use of the 24-h recall and thus we have stated the limitation in the use of the 24h recall under the discussion. Please see lines 527-532.

Point 8: Validation of the questionnaire used to obtain food craving, pica practices and food aversion?

Response 8: The data on the eating behaviour has been removed from the study.

The tables have been combined and restructured to report significant findings and this excluded the data on eating behaviour (food craving, pica practices, and food aversion).

Point 9: Was the data collection of anthropometric observed and verified by the investigators? How many people were involved in the measurement? How was it standardized to avoid inter-observer variations?

Response 9: The measurement of the mid-upper arm circumference was performed by two trained enumerators at a time. One investigator assisted and observed the measurement to reduce unforeseen errors. MUAC measurement was standardized by taking it twice for each participant, please see lines 214-215. We believe the training giving to the enumerators will reduce any inter-observer variations.

Point 10: BMI may not be the most accurate method for assessment of nutritional status in pregnancy. Weight gain during pregnancy is more accurate rather than BMI to assess nutritional status and fetal growth?

Response 10: BMI measurement has been removed from the study, as it was not significant in the study, and not a reliable nutritional status tool during pregnancy.

Point 11: May not be necessary to include all proportions; only those associations that are significant may be highlighted in the results. Avoid repetition of the results both in word and tables.

Response 11: The description of the results for the tables have been revised to report relevant and significant findings. Please see lines 288-352.

Point 12: Table 4 may be retained over the others. There are too many tables, the rest of the tables (1,2,3) may be combined to include only relevant findings.

Response 12: Tables 1,2,3 have been combined and revised to include only relevant findings. Please see new Tables 1-3.

Point 13: Although the discussion is well written, it is too lengthy as many theoretical explanations are given. Make the discussion crisper and concise.

Response 13: The discussion has been revised to make it concise. Please see lines 436-525.

Point 14: You can include the reasons as to what might explain the similarities and differences between the current study and the other studies used to compare the findings.

Response 14: The discussion on sociodemographics has been removed, because it does contribute much significant in the study. We also wanted to reduce the length of the discussion.

Point 15: Add a note on the limitations of the study, potential sources of bias, and the effort made, if any to address these biases.

Response 15: The limitations of the study have been included in the discussion. The study has limitations. The use of 24 h dietary recalls may lead to over-or under-estimations [Arsenault et al., 2020]. In the present study, we used a 3-day repeated recall to cover up for this. Also, a couple of food items consumed were not among the listed foods in the nutrient analysis template used to estimate nutrient levels. Instead, the template had similar foods that were used. This could have affected the estimated nutrient levels. Interpretation of the findings may not be generalized to the rest of Ghana, but probably indicate a similar situation of adolescents in other Regions of Ghana. We are aware of some bias in using a first come first served procedure in sampling. However, we employed convenience sampling by using a first come first served basis due to the low turnout of the pregnant teenagers at the clinics. As such, this sampling method was, therefore, appropriate to help obtain the study sample size and a representative population. Except for participants who were ineligible for this study, all participants who came and were eligible were recruited. Thus, there is less likely to have a probability bias risk involved in the sampling process. Please see lines 527-538.

Response to Reviewer Three Comments

Point 1: Describe the formula used to estimate the sample size of 420.

Response 1: The sample size was calculated using the formula from Charan and Biswas study [37]: 

n= 2(Z α/2 + Z β )2 p(1-p)/(P1-P2)^2. 

Where, n=sample size, Z α/2 =1.96 at type 1 error of 5%, Z β = 0.84 at 80% power, P1=LBW in pregnant adolescents with adequate nutritional status (11.6%), P2 = LBW in pregnant adolescents with poor nutritional status (23.3%), p1-p2= difference in prevalence of low birth weight between pregnant adolescents with adequate nutritional status at birth and those with inadequate nutritional status, and p= pooled prevalence= (p1 +p2)/2. A pilot study among pregnant adolescents in the area reported LBW prevalence of 23.3% [38]. Hence, we proposed that LBW in pregnant adolescents with adequate nutritional status would be 11.4%, while those with poor nutritional status would remain 23.3%, a reduction of just above half (51.1%). Hence, p1=11.4%, p2=23.3%, their proportions being p1=0.114 and p2=0.233, and p=(0.114 +0.233)/2= 0.1735.

Using the above descriptives, the sample size n= 2(1.96+0.84) 2 x 0.1735(1-0.233)/(0.114-0.233) 2, n=2.09/0.01, equal 209 was calculated, which implied we needed to recruit 209 participants in each arm of the study (half in the poorly nourished group and a half in the well-nourished group) making 418 participants showing a significant association between poor nutrition and LBW. However, we added 10% attrition to give 460 participants who were needed. However, the study had 416 participants. Please see lines 121-137.

Point 2. If Non-Parametric test has been used (Mann Whitney U test), then continuous variables may be expressed as Median and Interquartile range.

Response 2: The continuous variables expressed as means have been revised to the median and interquartile range. Please see Tables 1, 2, and 3.

Point 3. For variables such as mean MUAC, mean Serum PAB, mean Serum Ferritin, it seems that Independent t test is appropriate rather than Mann Whitney U test as mentioned in Table 2 and Table3.

Response 3. Tables 2 and 3 have been revised to use an independent t-test to analyze variables such as mean MUAC, mean Serum PAB, mean Serum Ferritin. Please see Tables 2 and 3.

Point 4. Mention the quality and adequacy of the model used.

Response 4: The binary regression model is an efficient way to determine the effect size of a predicting variable to the outcome variable. The adequacy of the data was described by the Hosmer-Lemeshow goodness of fit for the model. A p value > 0.05 shows the data are sufficient for the model, and in this study, we obtained p values > 0.05 for both models; anaemia model (p = 0.278) and depleted body iron stores model (p = 0.867). The overall predictive percentage was above 50 percent for the two models used, which also shows the quality of the data used in the regression model. The overall predicted percentage for the anaemia model was 76.3%, while that of the depleted body iron store model was 95.6%.

Response to Reviewer Four Comments

Reviewer #4 Point 1: IDA has been thoroughly researched and it has been found that the most common cause is nutritional; it was not clear the need as to why authors sought those particular factors for study. Study results were highlighted, however, discussion with known literature could be better. Limitations of the study and generalizability and feasibility were not discussed well. Few sentences could be phrased better.

Response 1: This study was birthed out from our previous retrospective study. In the previous study, which involved a secondary data analysis of anaemia trend in the district, we reported a higher prevalence of anaemia among pregnant teenagers than adults, and mean Hb levels of both teenagers and adults were found to be higher by the 36th week of pregnancy than at antenatal registration. We observed a positive correlation between Hb at antenatal registration and 36th-week gestation, indicating that pre-pregnancy Hb levels, largely determined anaemia occurrence during the latter stages of pregnancy [21]. In this follow up study, we sought to understand the associated factors of anaemia among teenagers the teenagers. This group is understudied, and this makes our study and its findings crucial. Against this backdrop, we decided to look into all the possible factors from literature; such as dietary intakes, poverty status, hunger status, and biochemical markers that contribute to anaemia, and if these factors exist among adolescents in Ghana living in rural and urban areas. Please see lines 508-518. The discussion has been revised to incorporate only known literature. Please see lines 436-525. The limitation of the study has been addressed. Please see lines 527-538. Grammatical errors and sentence phrases have been corrected and passed through an English grammar editing tool known as Grammerly.com.

---

## [Editor Report · Decision Letter 1]

30 Mar 2021

PONE-D-20-21782R1

Factors associated with Iron Deficiency Anaemia among pregnant teenagers in Kumasi, Ghana: A hospital-based prospective cohort study

PLOS ONE

Dear Dr. Annan,

Thank you for submitting your manuscript to PLOS ONE. After careful consideration, we feel that it has merit but does not fully meet PLOS ONE’s publication criteria as it currently stands. Therefore, we invite you to submit a revised version of the manuscript that addresses the points raised during the review process.

We look forward to receiving your revised manuscript.

Kind regards,

Leeberk Raja, MD

Academic Editor

PLOS ONE

Journal Requirements:

Additional Editor Comments (if provided):

Thank you for revising the manuscript. Your work has a significant merit and potential to be accepted for publication. However, the figure presented lacks few pre-requisites for a bar chart. It has been pointed out by a reviewer and even after that the revised figure does not have Y axis and there are errors. It's suggested that you can think of presenting the data in the figure in a different format and submit a revised version of the manuscript. We will re-assess and let you know the final decision on your paper.

---

## [Author Response · Author response to Decision Letter 1]

1 Apr 2021

Joerg Heber, Ph.D., 

The Editor-in-Chief,

PLoS One Corporation,

California, USA.

POINT-BY-POINT REVISIONS IN RESPONSE TO REVIEWER/EDITOR COMMENTS

Dear Editor-in-Chief,

Thanks for reviewing our manuscript.

In line with the reviewer’s comments, the following revisions have been made to the Manuscript ID: PONE-D-20-21782R1 entitled “Factors associated with Iron Deficiency Anaemia among pregnant teenagers in Ghana: A hospital-based prospective cohort study’ by Reginald Adjetey Annan, Linda Afriyie Gyimah, Charles Apprey, Anthony Kwaku Edusei, Odeafo Asamoah-Boakye, Linda Nana Esi Aduku, Wisdom Azanu, and Herman E. Lutterodt. Below is our submission, highlight in blue colour, and yellow in the revised manuscript.

Journal Requirement:

1. Please review your reference list to ensure that it is complete and correct. If you have cited papers that have been retracted, please include the rationale for doing so in the manuscript text, or remove these references and replace them with relevant current references. Any changes to the reference list should be mentioned in the rebuttal letter that accompanies your revised manuscript. If you need to cite a retracted article, indicate the article’s retracted status in the References list and also include a citation and full reference for the retraction notice

Response: The reference list has been revised and it is complete. All references are correctly cited and none have been retracted. However, we have added another reference in our discussion and made changes to the numbering of the references. Please see the highlighted yellow in the discussion and reference list as the new changes made. Below is the new cited reference: 

63. Gyimah LA, Annan RA, Apprey C, Edusei A, Aduku LNE, Asamoah-Boakye O, Azanu W. and Lutterodt H. Dietary diversity and its correlates among pregnant adolescent girls in Ghana. PLoS ONE. 2021; 16(3): e0247979. https://doi.org/10.1371/journal.pone.0247979.

Response to Academic Editor Comment

Point 1: However, the figure presented lacks few pre-requisites for a bar chart. It has been pointed out by a reviewer and even after that the revised figure does not have Y axis and there are errors. It's suggested that you can think of presenting the data in the figure in a different format and submit a revised version of the manuscript.

Response 1: Figure 1 has been revised to show Y axis and errors addressed. Also,the figure was passed through PACE for the correct ‘tif‘ figure format of Plos One. Please see figure 1 file.

Yours faithfully,

Dr. Reginald Adjetey Annan.

(Corresponding Author).

---

## [Editor Report · Decision Letter 2]

5 Apr 2021

Factors associated with Iron Deficiency Anaemia among pregnant teenagers in Kumasi, Ghana: A hospital-based prospective cohort study

PONE-D-20-21782R2

Dear Dr. Annan,

We’re pleased to inform you that your manuscript has been judged scientifically suitable for publication and will be formally accepted for publication once it meets all outstanding technical requirements.

Kind regards,

Leeberk Raja, MD

Academic Editor

PLOS ONE

---

## [Editor Report · Acceptance letter]

14 Apr 2021

PONE-D-20-21782R2 

Factors associated with Iron Deficiency Anaemia among pregnant teenagers in Ashanti Region, Ghana: A hospital-based prospective cohort study 

Dear Dr. Annan:

I'm pleased to inform you that your manuscript has been deemed suitable for publication in PLOS ONE. Congratulations! Your manuscript is now with our production department. 

Kind regards, 

on behalf of

Dr. Leeberk Raja 

Academic Editor

PLOS ONE